# Optimizing CNN-Based Diagnosis of Knee Osteoarthritis: Enhancing Model Accuracy with CleanLab Relabeling

**DOI:** 10.3390/diagnostics15111332

**Published:** 2025-05-26

**Authors:** Thomures Momenpour, Arafat Abu Mallouh

**Affiliations:** Department of Computer Science, Manhattan University, Riverdale, NY 10471, USA

**Keywords:** knee osteoarthritis, Kellgren–Lawrence, machine learning, CNN, transfer learning, Cleanlab, multi-class classification, EfficientNet-B5

## Abstract

**Background:** Knee Osteoarthritis (KOA) is a prevalent and debilitating joint disorder that significantly impacts quality of life, particularly in aging populations. Accurate and consistent classification of KOA severity, typically using the Kellgren-Lawrence (KL) grading system, is crucial for effective diagnosis, treatment planning, and monitoring disease progression. However, traditional KL grading is known for its inherent subjectivity and inter-rater variability, which underscores the pressing need for objective, automated, and reliable classification methods. **Methods:** This study investigates the performance of an EfficientNetB5 deep learning model, enhanced with transfer learning from the ImageNet dataset, for the task of classifying KOA severity into five distinct KL grades (0–4). We utilized a publicly available Kaggle dataset comprising 9786 knee X-ray images. A key aspect of our methodology was a comprehensive data-centric preprocessing pipeline, which involved an initial phase of outlier removal to reduce noise, followed by systematic label correction using the Cleanlab framework to identify and rectify potential inconsistencies within the original dataset labels. **Results:** The final EfficientNetB5 model, trained on the preprocessed and Cleanlab-remediated data, achieved an overall accuracy of 82.07% on the test set. This performance represents a significant improvement over previously reported benchmarks for five-class KOA classification on this dataset, such as ResNet-101 which achieved 69% accuracy. The substantial enhancement in model performance is primarily attributed to Cleanlab’s robust ability to detect and correct mislabeled instances, thereby improving the overall quality and reliability of the training data and enabling the model to better learn and capture complex radiographic patterns associated with KOA. Class-wise performance analysis indicated strong differentiation between healthy (KL Grade 0) and severe (KL Grade 4) cases. However, the “Doubtful” (KL Grade 1) class presented ongoing challenges, exhibiting lower recall and precision compared to other grades. When evaluated against other architectures like MobileNetV3 and Xception for multi-class tasks, our EfficientNetB5 demonstrated highly competitive results. **Conclusions:** The integration of an EfficientNetB5 model with a rigorous data-centric preprocessing approach, particularly Cleanlab-based label correction and outlier removal, provides a robust and significantly more accurate method for five-class KOA severity classification. While limitations in handling inherently ambiguous cases (such as KL Grade 1) and the small sample size for severe KOA warrant further investigation, this study demonstrates a promising pathway to enhance diagnostic precision. The developed pipeline shows considerable potential for future clinical applications, aiding in more objective and reliable KOA assessment.

## 1. Introduction

Knee osteoarthritis (KOA) is a progressive joint disorder characterized by the degeneration of cartilage, leading to chronic pain and reduced mobility. It affects nearly 40% of individuals over the age of 70 [1] and remains a leading cause of disability in older adults. The standard clinical method for assessing KOA severity is the Kellgren–Lawrence (KL) grading system, which categorizes radiographs into five grades (0–4) based on osteophyte formation, joint space narrowing, and other radiographic features [1], as illustrated in Figure 1. However, this KL grading process is inherently subjective. Inter-rater reliability studies often report a wide range of kappa (κ) values for KL grading, indicating variable agreement among radiologists, particularly for early or borderline grades [2]. This diagnostic inconsistency underscores the need for automated, reproducible KOA assessment tools.

Recent advances in machine learning (ML) and computer vision have shown promise in automating KOA classification using radiographic images. Deep learning models, particularly convolutional neural networks (CNNs), have demonstrated strong performance in medical image analysis by learning hierarchical visual patterns directly from raw pixel data [4].

Among these architectures, EfficientNet [5] has emerged as a high-performing model that scales network depth, width, and input resolution in a balanced way. Its B5 variant offers a favorable trade-off between accuracy and computational efficiency, making it well suited for medical imaging tasks. EfficientNetB5’s use of MBConv blocks, swish activation, and squeeze-and-excitation layers further enhances its capacity to detect nuanced structural changes in knee joints. The principle of compound scaling used in EfficientNet is illustrated in Figure 2.

To mitigate the challenge of limited labeled medical data, we incorporate transfer learning by fine-tuning a model pre-trained on the ImageNet dataset. This approach allows the network to reuse generalized visual features learned from millions of natural images, accelerating convergence and improving generalization for KOA classification [6].

Beyond model architecture, the quality of training data plays a critical role in performance. Label noise—common in medical datasets—can degrade accuracy and lead to poor generalization. To address this, we integrate CleanLab [7], a data-centric tool that detects and corrects potential label errors by leveraging the predictions of a reference model. CleanLab enhances dataset reliability and, consequently, the robustness of the trained classifier. The conceptual process by which Cleanlab improves dataset quality is illustrated in Figure 3.

### Contributions of This Paper

This paper presents a robust and data-aware KOA classification pipeline. Our main contributions are as follows:We develop a deep learning pipeline using EfficientNetB5 and transfer learning for five-class KOA classification.We improve the benchmark accuracy on the widely used Kaggle KOA dataset from 69% to 82.07%, and increase macro-averaged recall from 57.87% to 80.34%, establishing a new state of the art for this task.We assess the effectiveness of CleanLab in relabeling noisy medical datasets, demonstrating its substantial impact on classification performance.We conduct an ablation study comparing raw-labeled training and CleanLab-relabeled training.We apply Grad-CAM to visualize and interpret the model’s decision-making process, showing that it consistently focuses on medically relevant regions in correctly classified images.

## 2. Literature Review

Knee osteoarthritis (KOA) represents a significant global health concern, frequently leading to pain and disability, particularly among the aging population. The radiographic assessment of KOA severity, often guided by the Kellgren–Lawrence (KL) grading system, is a cornerstone of clinical diagnosis and management. However, this traditional approach is inherently subjective, susceptible to considerable inter-rater and intra-rater variability, which can impact treatment decisions and the comparability of research outcomes. To mitigate these limitations and enhance diagnostic precision, the field of medical imaging has witnessed a substantial shift towards leveraging artificial intelligence, particularly deep learning (DL) methodologies. These advanced computational techniques offer the potential to develop automated, objective, and highly efficient systems for KOA severity classification, thereby supporting clinicians and advancing research. This review critically examines the evolution of DL applications in KOA assessment, focusing on prominent model architectures, crucial methodological considerations such as data preprocessing, and the prevailing challenges and future directions that shape this rapidly advancing domain.

### 2.1. Evolution of Deep Learning Models for KOA Classification

The initial foray of deep learning into automated KOA assessment primarily involved the application of foundational convolutional neural network (CNN) architectures. These early studies aimed to demonstrate the feasibility of using data-driven approaches for interpreting radiographic images. A notable example is the work by [8], which utilized an extensive dataset from the Osteoarthritis Initiative (OAI), comprising over 40,000 radiographic images. Their CNN-based system, validated against assessments from musculoskeletal radiologists, achieved an average accuracy of 71% and an F1 score of 70%. While this pioneering effort underscored the significant potential of DL in this domain, its evaluation was primarily comparable to that of human readers and did not extensively benchmark against other emerging automated methodologies, leaving room for further comparative studies.

A significant leap in performance and efficiency was observed with the widespread adoption of transfer learning. This technique allows models pre-trained on vast, diverse image datasets like ImageNet to be repurposed and fine-tuned for more specialized tasks, such as medical image analysis. The utility of transfer learning stems from the ability of these pre-trained models to learn robust hierarchical feature representations from general images, which are often transferable to medical imaging tasks, thereby reducing the need for extremely large medical datasets and shortening training times [9]. This approach has become a standard practice in KOA research, enabling the development of more sophisticated and accurate classification systems.

Numerous studies have since validated the efficacy of transfer learning across a spectrum of established CNN architectures. For instance, researchers in [10] customized a DenseNet-201 model, applying it to a dataset of 5478 X-ray images for a binary classification task (differentiating normal knees from those with osteoarthritis). By implementing preprocessing techniques to address class imbalances, they reported an accuracy of 82.48%. In a broader comparative investigation, ref. [11] evaluated six different pre-trained models—VGG16, VGG19, ResNet101, MobileNetV2, InceptionResNetV2, and DenseNet121—using a Kaggle dataset of 9786 knee X-ray images. Their findings highlighted ResNet101 as particularly effective, delivering the highest classification accuracies for both binary tasks (with one configuration reaching 89% accuracy and 86% recall) and the more granular multi-class KL grading (achieving 69% accuracy and 67% recall for a 5-class structure). This work also critically emphasized the contribution of meticulous image preprocessing, including segmentation and equalization, to maximizing model performance.

Further expanding the comparative landscape, ref. [12] undertook an extensive evaluation of ResNet-50, Xception, VGG16, EfficientNetB0, and DenseNet201 across varied classification scenarios (2, 3, 4, and 5 KL grades). Their results indicated that DenseNet201, when optimized with the RMSprop algorithm, was particularly proficient in binary classification, achieving an accuracy of 87.7% and an F1-score of 87.2%. For the more challenging 5-class classification, the Xception model demonstrated notable success with 67.8% accuracy and an F1-score of 68.8%, illustrating that different architectures might be optimal for tasks of varying complexity. Concurrently, the study by [13], which introduced the “Osteo-NeT” system, involved fine-tuning a sequential CNN, VGG-16, and ResNet-50 on a dataset of 3836 images for binary classification. Among these, VGG-16 emerged as the top performer, achieving a testing accuracy of 92%, a result attributed in part to careful denoising and image enhancement preprocessing steps.

Alongside these explorations of well-established architectures, there has been a growing interest in more recent models designed for computational efficiency without significant compromise on accuracy, a crucial factor for potential clinical deployment. Ref. [14] successfully adapted the MobileNetV3 Large architecture, known for its suitability for resource-constrained environments, for binary KOA classification using a dataset of 3836 knee X-ray images, achieving an accuracy of 83%. This highlighted the viability of deploying DL solutions on devices with limited computational power. More recently, Singh et al. [15] leveraged the capabilities of EfficientNet B5, a state-of-the-art model that systematically scales network depth, width, and resolution. Applied to a 3-class KOA classification task (“Healthy”, “Moderate”, “Severe”) on a dataset of 1500 images, their approach, which combined transfer learning with strategic fine-tuning and custom layer additions, yielded an impressive classification accuracy of 97%. The collective findings from these diverse studies, summarized in Table 1, underscore a clear trend: leveraging sophisticated pre-trained deep learning models significantly advances KOA classification. However, it is also evident that the reported performance metrics can vary substantially, influenced by factors such as the number of target classes, the size and nature of the datasets, the specific model architecture chosen, and the preprocessing pipeline implemented.

### 2.2. Impact of Data Preprocessing, Augmentation, and Dataset Characteristics

The ultimate performance and reliability of deep learning models in medical imaging are inextricably linked to the quality of the input data and the rigor of the preprocessing strategies employed. Within the KOA literature, a variety of preprocessing steps are commonly reported. Image augmentation techniques, including random rotations, translations, scaling, and flipping, are frequently used to artificially enlarge the training dataset and introduce greater variability, which can help models generalize better and reduce overfitting, especially when dealing with limited dataset sizes as seen in [10]. Beyond general augmentation, more specialized preprocessing is often vital. This includes automatic image segmentation to isolate the knee joint or specific regions of interest (ROIs), thereby ensuring the model focuses on pertinent anatomical structures [11,12]. Contrast enhancement methods, such as Contrast Limited Adaptive Histogram Equalization (CLAHE), are also applied to improve the visibility of subtle radiographic features critical for accurate grading [12]. Furthermore, denoising algorithms and other image refinement processes are utilized to enhance image clarity and reduce artifacts that might otherwise mislead the learning process [13].

The datasets that form the basis of these KOA studies exhibit considerable diversity in both source and scale. Large-scale, publicly accessible repositories, most notably the Osteoarthritis Initiative (OAI) database [8,11] (though the latter also reports using Kaggle data for some experiments), provide tens of thousands of meticulously collected images with associated clinical data, enabling the development and validation of highly robust models. Other research efforts frequently utilize smaller, often publicly available datasets from platforms like Kaggle [16] as referenced by [10,11,12,13,14]. These datasets typically range from a few thousand to just under ten thousand images. While data augmentation can partially mitigate the limitations of smaller dataset sizes, the inherent variability in image acquisition protocols, patient populations, and, importantly, the potential for label noise in medical images remain significant and persistent challenges. The subjective nature of the KL grading system, even when applied by experts, can lead to inconsistencies in ground truth labels. It is noteworthy that while meticulous preprocessing is a recurrent theme in the surveyed literature, the systematic identification and automated correction of these potential label inconsistencies within KOA datasets has received comparatively less direct attention, representing a critical underexplored avenue for enhancing model performance and reliability.

### 2.3. Addressing Challenges and Future Directions in Automated KOA Assessment

Despite the considerable progress and promising results demonstrated, several multifaceted challenges continue to shape the landscape of DL applications in KOA classification. A primary concern, frequently cited in the broader medical AI literature, is the “black-box” nature of many deep learning models [9]. The lack of transparency in how these models arrive at their predictions can be a significant barrier to clinical adoption, where understanding the reasoning process is often crucial for trust and accountability. Consequently, enhancing model interpretability through techniques such as attention mapping (e.g., Gradient-weighted Class Activation Mapping, Grad-CAM), which can visualize the image regions most influential in a model’s decision-making process, is an active area of research. Such methods can not only foster greater clinical trust but also provide valuable tools for error analysis and model refinement.

The interconnected issues of data scarcity, patient privacy, and model generalization also loom large, particularly when the goal is to develop AI systems that perform reliably and equitably across diverse patient populations and varied clinical imaging protocols [9]. Federated learning (FL) has emerged as a particularly compelling paradigm to address these concerns, especially in the context of sensitive medical data [17]. FL facilitates collaborative model training across multiple institutions without requiring the sharing of raw patient data. Instead, models are trained locally at each institution, and only anonymized model updates or parameters are aggregated centrally to create an improved global model. This iterative process allows for the development of more robust and generalizable models by leveraging larger and more diverse datasets while upholding stringent patient privacy standards. Although FL has shown significant potential in various medical imaging applications, its specific deployment and demonstrated advantages for complex multi-class KOA severity classification are still in the nascent stages of exploration, representing a significant and promising direction for future research endeavors.

Furthermore, the substantial computational resources often required for training state-of-the-art deep learning models can pose a practical barrier to their widespread development and deployment, particularly in settings with limited infrastructure [9]. This reality has catalyzed significant research into computationally efficient yet powerful architectures. Models such as MobileNetV3 [14] and the EfficientNet family [15] are designed to offer a favorable balance between predictive performance and resource consumption, thereby making sophisticated AI tools more accessible and feasible for integration into routine clinical workflows.

### 2.4. Synthesis and Identified Research Gap

The collective body of research unequivocally demonstrates that deep learning, particularly through the application of transfer learning techniques, holds immense promise for revolutionizing KOA severity classification. Various sophisticated architectures have shown the capability to achieve high levels of accuracy in both binary (normal vs. OA) and, albeit with greater difficulty, multi-class grading tasks as summarized in Table 1. The critical role of comprehensive preprocessing and data augmentation strategies in optimizing model performance is also a consistent finding across numerous studies. However, a thorough review of the existing literature also illuminates several important avenues that warrant further, more focused investigation.

Firstly, while the clinical standard for KOA assessment, the Kellgren–Lawrence grading system, defines five distinct severity classes, a considerable portion of the research simplifies the classification problem to binary or three-class scenarios. Robust and highly accurate five-class classification, which provides the most granular assessment, remains a more formidable challenge, and reported performance in this context generally lags behind simpler tasks. Achieving higher accuracy in five-class grading is crucial for closer alignment with clinical practice and for enabling more nuanced patient stratification.

Secondly, and critically, despite the well-documented subjectivity inherent in manual KL grading and the consequent high likelihood of label noise and inconsistencies within KOA datasets, few studies in the reviewed literature explicitly detail or implement methods for systematic label quality assessment and automated correction beyond standard image preprocessing. The potential impact of advanced tools specifically designed for identifying and rectifying label errors—such as Cleanlab—on the performance of state-of-the-art deep learning models, especially in the demanding context of 5-class KOA grading, has not yet been comprehensively explored. This oversight is significant, as noisy labels can substantially impair model training, leading to suboptimal performance and reduced reliability.

Therefore, this study is designed to address these specific gaps. We aim to rigorously investigate the performance of the EfficientNetB5 architecture, a model recognized for its excellent balance of high accuracy and computational efficiency, for the challenging task of five-class KOA severity classification based on the KL grading system. A central and novel aspect of our proposed methodology is the systematic integration of Cleanlab as a data-centric AI technique to preprocess and relabel the dataset. By explicitly targeting and mitigating potential label inconsistencies, we hypothesize that we can significantly enhance the robustness of the model training process, leading to improved classification accuracy and greater reliability of the resulting diagnostic tool. Through this focused approach—combining an advanced model architecture with sophisticated label correction for a clinically relevant multi-class problem—we seek to contribute to the development of more accurate, trustworthy, and clinically applicable automated systems for KOA assessment, ultimately aiming to improve diagnostic consistency and support better patient management.

## 3. Methodology

This study proposes a multi-stage deep learning approach for the classification of knee osteoarthritis (KOA) severity into five Kellgren–Lawrence (KL) grades (0: Healthy, 1: Doubtful, 2: Minimal, 3: Moderate, 4: Severe). The methodology encompasses dataset acquisition and detailed characterization including initial data splits, a two-phase preprocessing pipeline involving outlier removal and label correction with Cleanlab, followed by the training and evaluation of an EfficientNetB5 model using a weighted loss function to address class imbalance. An overview of the proposed approach is depicted in Figure 4.

### 3.1. Dataset Acquisition and Characterization

The primary dataset utilized in this research is sourced from Kaggle [16]. This dataset initially comprised radiographic knee X-ray images categorized into the five KL grades. The dataset was pre-split into training, validation, and testing sets. The initial distribution of images per KL grade within each set, prior to any preprocessing, was as follows:**Training set** (total original examples: 5780 images, with 4624 (80%) used after initial outlier check for Cleanlab input.–Grade 0 (Healthy): 2286 images–Grade 1 (Doubtful): 1046 images–Grade 2 (Minimal): 1516 images–Grade 3 (Moderate): 757 images–Grade 4 (Severe): 173 images**Validation Set** (Total examples: 826 images)–Grade 0 (Healthy): 328 images–Grade 1 (Doubtful): 153 images–Grade 2 (Minimal): 212 images–Grade 3 (Moderate): 106 images–Grade 4 (Severe): 27 images**Test Set** (Total examples: 1656 images)–Grade 0 (Healthy): 639 images–Grade 1 (Doubtful): 296 images–Grade 2 (Minimal): 447 images–Grade 3 (Moderate): 223 images–Grade 4 (Severe): 51 images

This distribution highlights a significant class imbalance across all sets, with Grade 0 being the most represented and Grade 4 the least. A preliminary manual review of a subset of images also suggested potential labeling inconsistencies, particularly for KL Grade 1 (Doubtful), which represents a subtle transitional stage. These observations motivated a robust data quality enhancement strategy involving outlier removal and systematic label correction, alongside techniques to handle class imbalance during model training. Figure 5 illustrates sample images from each KL grade within the dataset.

### 3.2. Data Preprocessing

A two-phase preprocessing strategy was implemented to enhance data quality prior to final model training: (1) outlier removal using Alibi Detect, and (2) label correction using Cleanlab. All images were resized to 224×224 pixels for model input.

#### 3.2.1. Phase 1: Outlier Detection and Removal

To reduce noise and remove anomalous samples that could negatively impact model training, an outlier detection step was performed using the Alibi Detect Python (version 0.12.1) library [18]. Specifically, an Autoencoder-based outlier detector was employed. An Autoencoder is trained to reconstruct its input data; images that are significantly different from the majority distribution will result in higher reconstruction errors. This reconstruction error serves as an outlier score for each image. A threshold of 0.8 was established based on empirical evaluation; any image with a predicted outlier score greater than or equal to this threshold was considered an outlier and removed from the dataset. This process was applied exclusively to the training set to help CleanLab more accurately detect labeling errors. By removing label noise beforehand, we reduced the risk of inconsistent annotations misleading the relabeling process.

#### 3.2.2. Phase 2: Label Correction with Cleanlab

Following outlier removal, Cleanlab [7] was utilized to identify and correct potentially mislabeled images within all three data splits (training, validation, and testing). This step was motivated by the observation of inter-grade subtleties and aimed to improve the overall label quality. The rationale for applying label correction to validation and test sets was to ensure that these sets, used for model selection and final evaluation respectively, also benefit from the improved label quality, providing a more reliable assessment of the model’s performance on data presumed to be more accurately labeled according to a consistent, data-driven criterion.

The Cleanlab process involved the following steps:**Model for Generating Inputs to Cleanlab:** An initial EfficientNetB5 model (hereafter referred to as the “Cleanlab-input model”) was trained on the outlier-removed training data. This model architecture is detailed further in Section 3.3 but was trained using standard categorical cross-entropy without class weights for this specific preliminary stage. It was fine-tuned with dropout, batch normalization, and kernel regularizers.**Generating Predictions and Embeddings:** The trained Cleanlab-input model was used to generate out-of-sample predicted class probabilities and feature embeddings for all images in the (outlier-removed) training, validation, and test sets. Feature embeddings were extracted from the penultimate layer of this model. For generating these inputs, data generators were configured with ‘shuffle=False’ to maintain a one-to-one correspondence between predictions/embeddings and the original image files.**Identifying Label Issues:** The Datalab object from Cleanlab was initialized with the respective dataset partitions and their original labels. The find_issues method was then invoked, utilizing both the generated predicted probabilities and the feature embeddings as input.**Relabeling Criterion and Implementation:** Cleanlab’s get_issues(“label”) method provided a list of samples identified as potential label issues, along with a ‘predicted_label’ (Cleanlab’s suggested corrected label) for each. An image’s label was updated if its original ‘given_label’ differed from Cleanlab’s ‘predicted_label’. The relabeling was physically implemented by moving the image file to the directory corresponding to the Cleanlab-suggested class label.

This relabeling process resulted in a notable number of label changes. Specifically, 666 images were relabeled in the training set (out of 5778 examples), 215 in the validation set (out of 826 examples), and 366 in the test set (out of 1656 examples). Common reassignments in the training set included 176 images moving from original Class 1 (Doubtful) to new Class 0 (Healthy), 116 images from original Class 0 to new Class 1, and 100 images from original Class 2 (Minimal) to new Class 1. Similar re-classifications occurred in the validation and test sets. Detailed confusion matrices illustrating these label changes for each dataset split are presented later in the Results section. This systematic relabeling aimed to create a more consistent and accurately labeled dataset across all partitions.

### 3.3. Model Architecture: EfficientNetB5 with Custom Classifier Head

The core architecture for KOA severity classification is EfficientNetB5 [5], selected for its demonstrated high accuracy and computational efficiency. We employed transfer learning, initializing the EfficientNetB5 backbone with weights pre-trained on the extensive ImageNet dataset [19]. The original top classification layers of the pre-trained EfficientNetB5 were removed and replaced with a custom classifier head specifically designed for the 5-class KOA task. This custom head, as illustrated in Figure 6, comprises the following layers in sequence:A Global Average Pooling 2D (GlobalAveragePooling2D) layer, which reduces the spatial dimensions of the feature maps from the EfficientNetB5 base.A dropout layer with a rate of 0.5, introduced to mitigate overfitting by randomly deactivating neurons during training.A Dense (fully connected) layer with 256 units, employing a Rectified Linear Unit (ReLU) activation function. $L_{2}$ kernel regularization was applied to this layer to further penalize large weights and prevent overfitting.A batch normalization layer, applied after the dense layer and before the subsequent dropout, to stabilize the training process and improve generalization by normalizing the activations.A second dropout layer with a rate of 0.4, providing an additional stage of regularization.A final Dense output layer with 5 units, corresponding to the five KL grades. This layer uses a Softmax activation function to produce a probability distribution over the classes.

### 3.4. Final Model Training and Optimization

The final EfficientNetB5 model, incorporating the custom classifier head, was trained on the Cleanlab-remediated training dataset and validated using the Cleanlab-remediated validation dataset.

**Data Augmentation:** To enhance model robustness and reduce the risk of overfitting to the training data, on-the-fly data augmentation was applied during training. This included random horizontal flipping of images and random rotations (e.g., within a range of ±10 degrees).**Optimizer:** The Adam optimizer [20] was utilized for its adaptive learning rate capabilities and proven efficiency in deep learning applications [21].**Learning Rate:** A base learning rate (η) of $1\times 10^{-4}$ was set for the Adam optimizer. This relatively small learning rate encourages more stable convergence, which is particularly important when fine-tuning pre-trained models [22,23]. The conceptual importance of learning rate selection is illustrated in Figure 7.**Loss Function and Class Weighting:** Categorical cross-entropy was employed as the loss function. To address the inherent class imbalance in the KOA dataset, class weights were incorporated into the loss function during training. These weights were calculated as inversely proportional to the class frequencies observed in the Cleanlab-remediated training set. This strategy assigns a higher penalty to misclassifications of underrepresented classes (e.g., KL Grade 4), thereby encouraging the model to learn more effectively from these minority classes.**Batch Size:** The batch size for training was dynamically determined based on the length of the training dataset, with an upper limit of 80 samples per batch.**Epochs and Early Stopping:** The model was set to train for a maximum of 50 epochs. An early stopping mechanism was implemented, monitoring the validation loss at the end of each epoch. If the validation loss did not show improvement for three consecutive epochs (patience = 3), the learning rate was decayed (e.g., by a factor of 0.2). Training was terminated if the learning rate fell to a minimum threshold of $1\times 10^{-6}$ without any further improvement in validation loss, or if the 50-epoch limit was reached [24,25].

**Figure 7 diagnostics-15-01332-f007:**
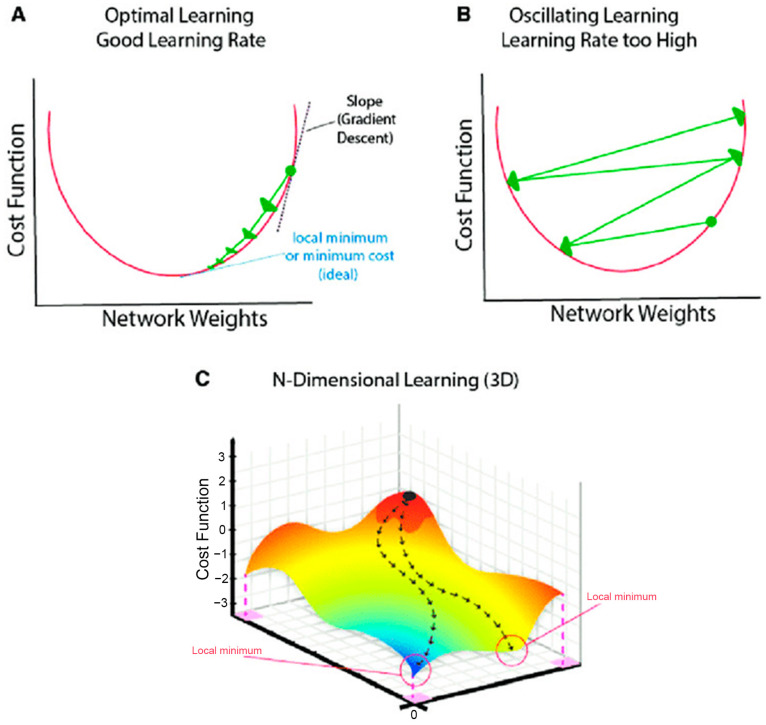
Conceptual illustration of learning rate effects on optimization: (**A**) An optimal learning rate facilitates smooth and stable convergence. (**B**) An excessively high learning rate can cause overshooting. (**C**) A 3D visualization of a cost surface illustrates learning rate influence. (Reproduced from [26]).

### 3.5. Performance Evaluation Metrics

The performance of the finally trained and optimized KOA classification model was rigorously evaluated on the Cleanlab-remediated test set using the following standard metrics:**Accuracy:** The proportion of all predictions that were correct.$$\mathrm{Accuracy}=\frac{\mathrm{TP}+\mathrm{TN}}{\mathrm{TP}+\mathrm{TN}+\mathrm{FP}+\mathrm{FN}}$$**Precision:** For each class, the proportion of true positive predictions among all instances predicted as that class.$$\mathrm{Precision}=\frac{\mathrm{TP}}{\mathrm{TP}+\mathrm{FP}}$$**Recall (Sensitivity):** For each class, the proportion of true positive predictions among all actual instances of that class.$$\mathrm{Recall}=\frac{\mathrm{TP}}{\mathrm{TP}+\mathrm{FN}}$$**F1-score:** The harmonic mean of Precision and Recall, calculated for each class.$$F1-\mathrm{score}=2\cdot \frac{\mathrm{Precision}\cdot \mathrm{Recall}}{\mathrm{Precision}+\mathrm{Recall}}$$

In these formulas, TP, TN, FP, and FN represent True Positives, True Negatives, False Positives, and False Negatives, respectively. For multi-class evaluation, these metrics are typically calculated on a per-class basis and then averaged (e.g., macro-average or weighted-average).

### 3.6. Computational Complexity and Inference Efficiency

EfficientNetB5 is attractive in clinical settings because it delivers high accuracy with a moderate computational footprint. Using the keras_flops utility, our final five-class model contains 29.05 million parameters in total, of which 28.87 million are trainable. The same tool reports an inference cost of only 4.82 GFLOPs for a 224×224 RGB image.

On an NVIDIA (Headquarter is in Santa Clara, CA, USA) RTX 2080 Ti, the average forward-pass latency is approximately 12 ms per image (batch size = 32), while on an Intel (Headquarter is in Santa Clara, CA, USA) i7-10700 CPU it is around 85 ms per image. For comparison, the canonical ResNet-101 architecture contains 44.5 million parameters and approximately 7.6 GFLOPs, whereas MobileNetV3-Large has 5.4 million parameters and roughly 0.22 GFLOPs, according to their original specifications [27,28].

Thus, our model is markedly lighter than high-capacity backbones like ResNet-101, yet substantially more accurate than lightweight baselines such as MobileNetV3 previously applied to this dataset. This balance of performance and efficiency supports the feasibility of near real-time deployment in clinical environments.

## 4. Results

This section presents the performance of our proposed EfficientNetB5 model, enhanced with outlier removal and Cleanlab-based label correction, for the five-class classification of knee osteoarthritis (KOA) severity. We detail the overall model performance, training dynamics, the impact of our data-centric preprocessing pipeline including statistical significance, class-specific results, model interpretability using Grad-CAM, and a comparison with existing studies.

### 4.1. Overall Model Performance

On the Cleanlab-remediated test set, the final EfficientNetB5 model (trained on the Cleanlab-remediated training data) achieved an overall accuracy of 82.07%. The comprehensive performance, including precision, recall, and F1-score for each class, along with macro and weighted averages, is detailed in Table 2. The weighted average F1-score was 0.8154, and the macro average F1-score was 0.8019, indicating robust performance across the different KL grades. Figure 8 presents the confusion matrix for this model on the test set, illustrating its predictions against the Cleanlab-remediated true labels for each class. Each row represents the instances in an actual (true) class, while each column represents the instances in a predicted class. Values on the diagonal indicate correctly classified instances (True Positives for that class), while off-diagonal values represent misclassifications (e.g., a value in row ‘Actual Class A’, column ‘Predicted Class B’ indicates instances of Class A misclassified as Class B).

### 4.2. Training Dynamics

The final model was trained for a total of 13 epochs before early stopping criteria were met, taking approximately 51 min. Figure 9 displays the training and validation accuracy and loss curves over the epochs for this model. The lowest validation loss of 0.67 was observed at epoch 13, while the highest validation accuracy of 82.00% was achieved at epoch 9. These curves demonstrate stable learning without significant overfitting.

### 4.3. Impact of Data-Centric Preprocessing: Ablation Study

To quantify the contribution of our full data-centric pipeline, which includes outlier removal and Cleanlab-based label correction, an ablation study was conducted. We compared the performance of our final EfficientNetB5 model (trained on the fully preprocessed and Cleanlab-remediated training dataset) against a baseline EfficientNetB5 model (trained on the raw training dataset with only standard data augmentation). Performance metrics are reported in Table 3. The “Baseline” configuration was evaluated on the original test set labels, while the “Final Pipeline” was evaluated on the Cleanlab-remediated test set labels.

The results in Table 3 clearly demonstrate the substantial gains achieved with our full data-centric pipeline. Accuracy improved from 66.36% (baseline) to 82.07% (final pipeline), and the Macro F1-score increased significantly from 60.23% to 80.19% when each model was evaluated on its respective optimally prepared test set.

To further probe the statistical significance of these improvements and understand the influence of label quality on evaluation, McNemar’s test was performed to compare the predictions of the baseline model against our final Cleanlab-pipeline model under two test set conditions. The final model (accuracy 82.07%) when evaluated on CleanLab-remediated test labels showed a statistically significant improvement compared to the baseline model when also evaluated on these same CleanLab-remediated test labels ($p\approx 3.37\times 10^{-14}$, i.e., p<0.00001). However, when both models were evaluated on the original (potentially noisy) test set labels, the difference in their accuracies was not statistically significant (p=0.076). This suggests that while our pipeline significantly improves model performance on data presumed to be more accurately labeled by Cleanlab, evaluation on the original noisy labels may obscure the true extent of this improvement due to inconsistencies within those original labels. These findings underscore the critical importance of dataset quality and robust labeling for both effective model training and reliable evaluation, though additional validation (e.g., expert annotation) is needed to confirm the definitive correctness of the relabels.

### 4.4. Cleanlab Relabeling Analysis

The Cleanlab process led to considerable label adjustments across all dataset splits, highlighting its role in refining data quality. Figure 10 illustrates these changes via heatmaps for the training, validation, and testing datasets. Across all datasets, KL Grade 1 (Doubtful) exhibited the highest number of relabeling corrections. For instance, in the training dataset (which had 732 relabeled images out of 5778 after outlier removal), common transitions for original Grade 1 images were to Grade 0 (176 instances) and Grade 2 (88 instances). This pattern suggests initial ambiguity or misclassification between healthy, doubtful, and minimally affected knees. Similar tendencies were observed in the validation set (214 images relabeled) and the test set (364 images relabeled). Overall, the lower-severity grades (0, 1, and 2) underwent more extensive relabeling compared to the higher-severity grades (3 and 4), whose original labels appeared more stable, likely due to more distinct radiographic features.

### 4.5. Class-Wise Performance Analysis

A detailed look at the per-class performance of our final model (from Table 2) reveals further insights:**Class 0 (Healthy):** The model demonstrated excellent performance, achieving high precision (0.8492) and recall (0.9369). This indicates strong efficacy in correctly identifying healthy knees.**Class 1 (Doubtful):** This class proved most challenging, with the lowest precision (0.6286) and recall (0.5176). This aligns with the subjective nature of this grade and the significant relabeling activity observed by Cleanlab. Notably, Cleanlab’s intervention substantially improved recall for this class from an initial 0.28 (baseline model on original labels) to 0.5176 (final model on remediated labels).**Classes 2-3 (Minimal, Moderate):** The model performed robustly for these intermediate grades, with F1-scores of 0.8068 (Class 2) and 0.8756 (Class 3), indicating good capability in distinguishing these varying severities.**Class 4 (Severe):** Despite having the smallest support (48 samples in the remediated test set), the model achieved very high recall (0.8958) and precision (0.8431) for severe KOA cases. This is clinically important for identifying patients requiring urgent attention.

### 4.6. Model Interpretability with Grad-CAM

To understand the model’s decision-making process, Gradient-weighted Class Activation Mapping (Grad-CAM) was employed. Figure 11 displays Grad-CAM heatmaps for correctly predicted samples from each KL grade by our final model. The highlighted regions consistently align with clinically relevant areas of the knee joint, such as regions of joint space narrowing, osteophyte formation, and subchondral bone changes. This suggests the model learns medically pertinent features for classification.

Figure 12 shows a Grad-CAM visualization for a misclassified instance (True Grade 2, Predicted Grade 1). While the model focuses on the joint region, the attention map is more diffuse compared to correctly classified cases. This may indicate model uncertainty when faced with subtle or borderline degenerative features, reinforcing the challenges associated with these ambiguous cases. Such interpretability tools are valuable for building trust and understanding model behavior in clinical contexts.

### 4.7. Comparison with Previous Studies

Our approach, combining EfficientNetB5 with transfer learning and our full data-centric preprocessing pipeline (outlier removal and Cleanlab-based relabeling), was compared against previously reported models for KOA classification, particularly those evaluated on the same Kaggle dataset [16] for a five-class task. As shown in Table 4, our method achieves an accuracy of 82.07% on the Cleanlab-remediated test set, which surpasses the highest previously reported accuracy of 69% by ResNet-101 [11] for the 5-class problem on this dataset.

While some studies, such as [14] (reporting 83% accuracy with MobileNetV3), achieved numerically similar or higher accuracies, they were typically evaluated on simplified binary (KOA vs. Normal) classification tasks. Our model addresses the more granular and clinically relevant five-class Kellgren–Lawrence grading. The substantial improvement over other five-class models using the same base dataset underscores the efficacy of our comprehensive data-centric pipeline. Furthermore, our model’s strong macro-averaged F1-score (80.19%) and recall (80.34%) suggest a balanced sensitivity across all severity levels. The performance gain from 66.36% (baseline EfficientNetB5 on original data) to 82.07% (final pipeline on Cleanlab-remediated data) further highlights the critical importance of addressing data quality issues like outliers and label noise for achieving reliable and high-performing classification models.

## 5. Discussion and Conclusions

This study demonstrates the efficacy of a fine-tuned EfficientNetB5 model in grading knee osteoarthritis (KOA) from X-ray images, achieving an overall accuracy of 82.07% and a macro average recall of 80.34%. This result establishes a new state-of-the-art benchmark for academic research utilizing the specified Kaggle dataset [16] for five-class KOA severity classification, underscoring the significant potential of advanced deep learning methodologies in medical image analysis.

### 5.1. Model Performance, Data-Centric Impact, and Clinical Implications

The EfficientNetB5 model, enhanced by our data-centric pipeline, exhibited strong performance, particularly in identifying healthy knees (KL Grade 0) and those with advanced osteoarthritis (KL Grade 4). This robust differentiation is crucial for clinical decision-making, offering a valuable assistive tool for radiologists. It has the potential to streamline diagnostic and grading phases, thereby enhancing the efficiency and consistency of clinical workflows, and potentially reducing inter-observer variability.

A critical factor in achieving these results was the systematic approach to data quality improvement. The implementation of Cleanlab for label correction directly addressed the prevalent challenge of label noise and inconsistencies within the Kaggle dataset used for this study. As evidenced by our ablation study and statistical analysis (p<0.00001 on remediated labels), correcting mislabeled instances significantly improved the quality of the training data. This, in turn, enabled the EfficientNetB5 architecture to better discern the complex radiographic patterns associated with KOA, leading to the noteworthy improvements in overall accuracy and class-specific recall, especially for the ambiguous Class 1. The Grad-CAM visualizations further supported these quantitative findings by indicating that the model learned to focus on clinically relevant features.

### 5.2. Challenges and Limitations

Despite the significant advancements, this study acknowledges certain challenges and limitations. The model, while improved, still faced difficulties in the precise classification of Class 1 (Doubtful) cases, which represent borderline or very early-stage osteoarthritis. This underscores the inherent complexity of detecting subtle radiographic changes and the potential for continued ambiguity even after data-driven label correction. While Cleanlab improved recall for this class from 0.28 to 0.5176, its precision (0.6286) indicates that distinguishing this class perfectly remains an open challenge.

Furthermore, this study was conducted using a single, albeit commonly referenced, Kaggle dataset. While this allowed for comparison with the existing literature, the findings’ generalizability to different patient populations, imaging equipment, and clinical settings requires further investigation through external validation on diverse datasets. The “true” accuracy of all labels corrected by Cleanlab, while showing improved model performance, was not independently verified by a panel of radiologists, which remains a gold standard for label validation. Lastly, although the model performed well for severe cases (Class 4), the relatively small sample size for this class warrants caution when extrapolating its performance to all instances of severe KOA.

### 5.3. Future Research Directions

To build upon the current findings and address the identified limitations, future research should be directed towards several key areas:**Enhanced Dataset Curation and Validation:** Further refinement of datasets, particularly for ambiguous Class 1 cases, is essential. This should involve multi-reader expert reviews to establish a consensus gold standard for borderline KOA. Additionally, validating the current model and future iterations on large-scale, multi-center external datasets (such as the Osteoarthritis Initiative (OAI) [29] or other clinical repositories) is a critical next step to confirm generalizability.**Advanced Model Architectures and Strategies:** Explore sophisticated ensemble methods or multi-stage classification systems specifically designed to improve accuracy for challenging, ambiguous cases. For instance, a hierarchical approach could first distinguish healthy from osteoarthritic knees, followed by a dedicated model for fine-grained severity grading. Investigating 3D convolutional neural networks, if multiple X-ray views or volumetric data (e.g., CT and MRI) are considered, could also leverage richer spatial information [30].**Broader Clinical Validation and Utility Studies:** Conduct comprehensive clinical validation studies to assess the model’s performance and utility in real-world radiological practice. This includes evaluating its impact on diagnostic agreement among radiologists and its potential to predict osteoarthritis progression over time using longitudinal data [31].**Integration of Multimodal Data:** Future iterations could benefit from incorporating additional data sources, such as patient demographics, clinical symptoms, serological markers, or other imaging modalities (e.g., MRI), to develop a more holistic and potentially more accurate diagnostic and prognostic tool for KOA.**Federated Learning for Collaborative Research:** Explore federated learning approaches to train more robust and generalizable models on diverse datasets from multiple institutions without compromising patient data privacy, which is particularly relevant for medical imaging tasks.

### 5.4. Overall Impact and Outlook

This study contributes significantly to the automated classification of knee osteoarthritis, establishing a new performance benchmark on the specified Kaggle dataset through a rigorous data-centric deep learning approach. The evident impact of addressing label noise with techniques like Cleanlab highlights a crucial direction for improving medical image analysis pipelines. By demonstrating a pathway to enhanced accuracy, this research provides a strong foundation for future advancements in AI-assisted KOA diagnosis.

As these AI technologies continue to mature, their potential to streamline clinical workflows, facilitate earlier and more objective KOA diagnosis, and ultimately improve patient outcomes becomes increasingly tangible. The integration of reliable AI tools into radiological practice could lead to more standardized assessments of osteoarthritis severity, supporting personalized treatment strategies and the more effective monitoring of disease progression.

In conclusion, while this research marks a significant step forward in automated KOA grading, it also reinforces the ongoing need for interdisciplinary collaboration. Continued efforts involving computer scientists, radiologists, and clinicians are essential to translate these technological capabilities into robust, trustworthy, and clinically integrated solutions that can meaningfully improve musculoskeletal healthcare. 

## Figures and Tables

**Figure 1 diagnostics-15-01332-f001:**
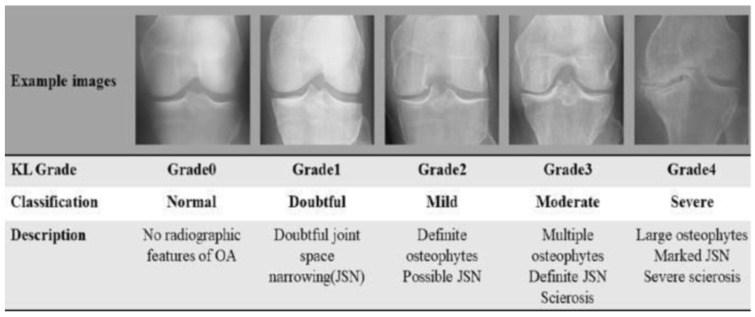
The Kellgren–Lawrence grading system is widely used to classify knee osteoarthritis severity based on radiographic features [3].

**Figure 2 diagnostics-15-01332-f002:**
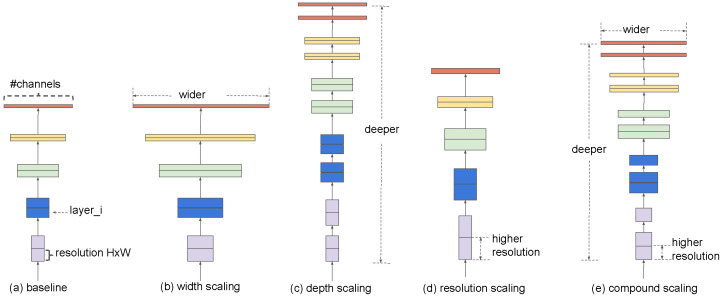
Compound scaling in EfficientNetB5 balances depth, width, and resolution to improve accuracy without significantly increasing complexity [5].

**Figure 3 diagnostics-15-01332-f003:**
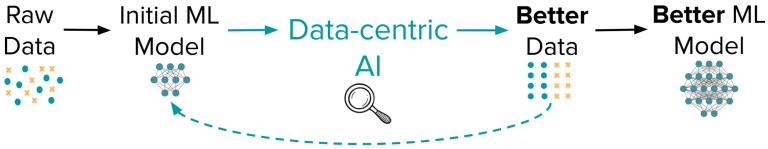
CleanLab (version 2.7.0) improves dataset quality by identifying mislabeled samples and refining noisy training data [7].

**Figure 4 diagnostics-15-01332-f004:**
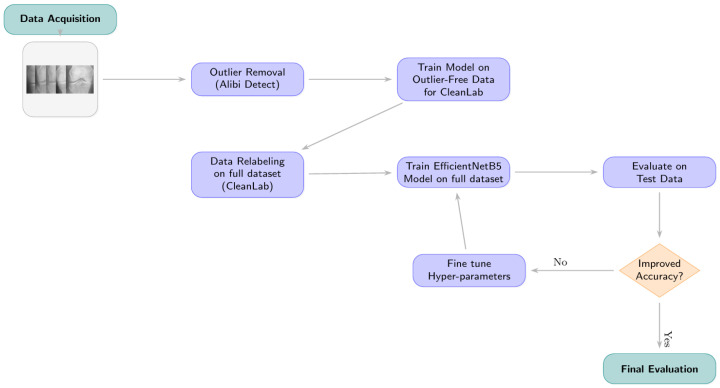
Proposed approach for KOA detection and classification, detailing data acquisition, preprocessing with outlier removal and Cleanlab-based relabeling, model training with class weighting, and evaluation.

**Figure 5 diagnostics-15-01332-f005:**
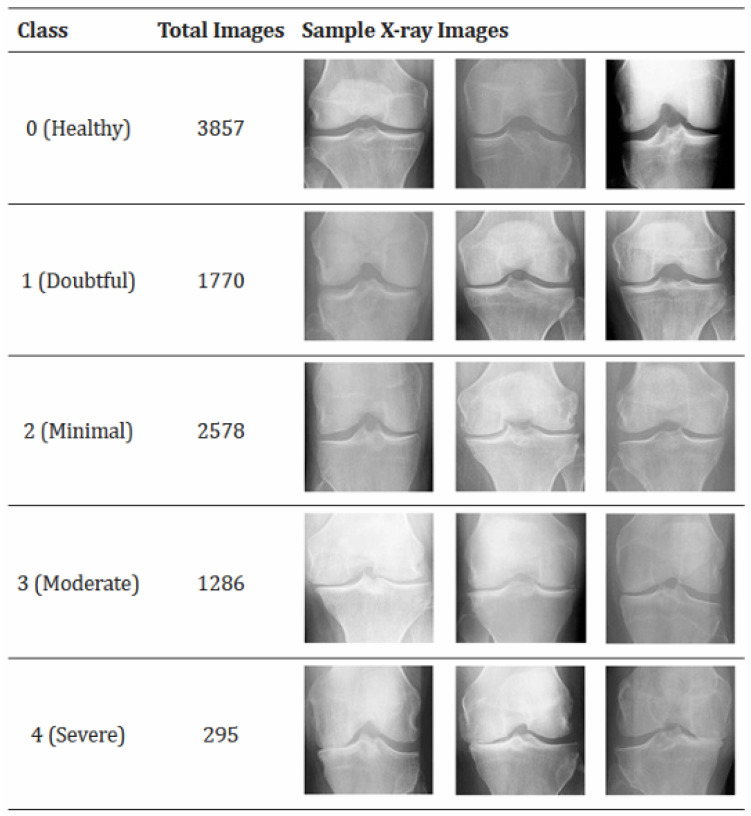
Sample radiographic images from the dataset illustrating the five Kellgren–Lawrence grades of knee osteoarthritis. (Original diagram credited to [11], adapted to represent samples from the current study’s dataset).

**Figure 6 diagnostics-15-01332-f006:**
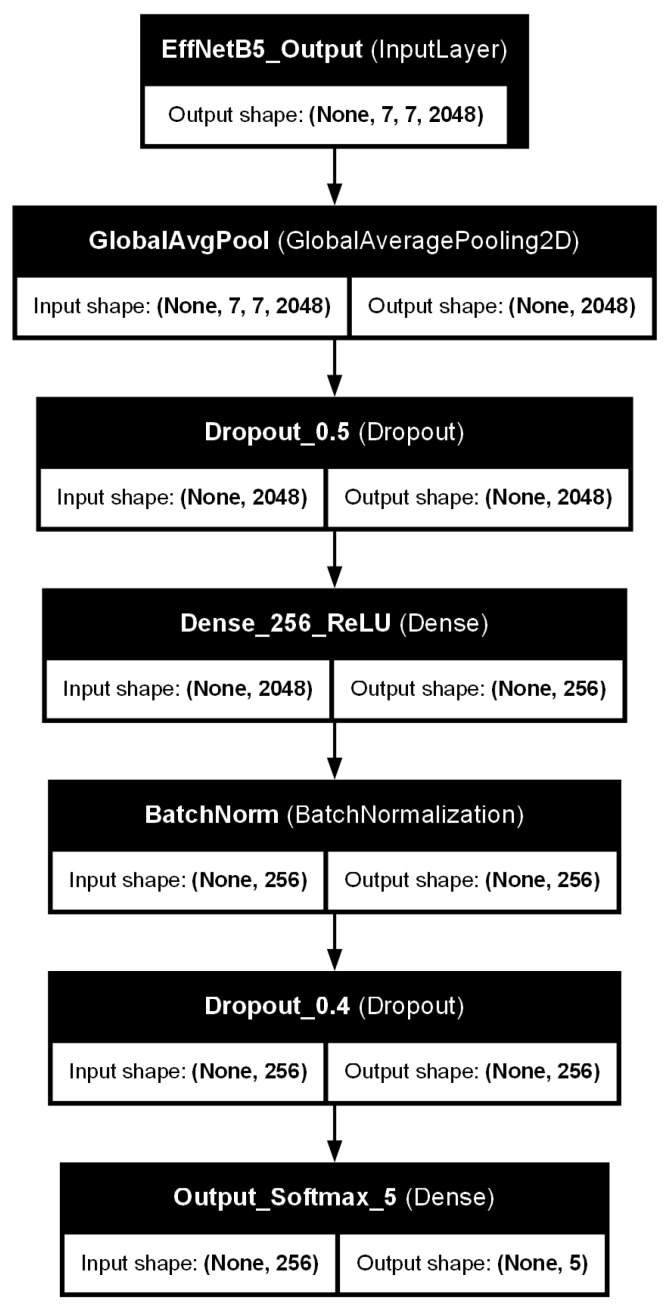
Architecture of the custom classifier head appended to the pre-trained EfficientNetB5 base model. It includes global average pooling, dropout layers, a ReLU-activated dense layer with batch normalization and $L_{2}$ regularization, and a final softmax layer for 5-class KOA classification.

**Figure 8 diagnostics-15-01332-f008:**
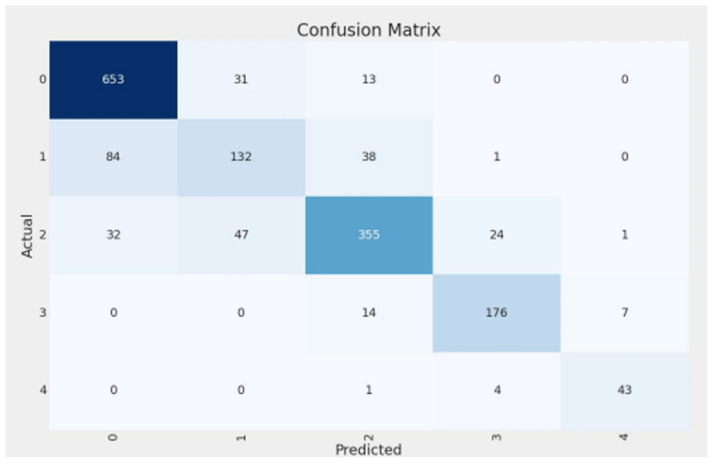
Confusion matrix for the final EfficientNetB5 model on the Cleanlab-remediated test set, showing predictions versus true labels for the five KOA severity classes.

**Figure 9 diagnostics-15-01332-f009:**
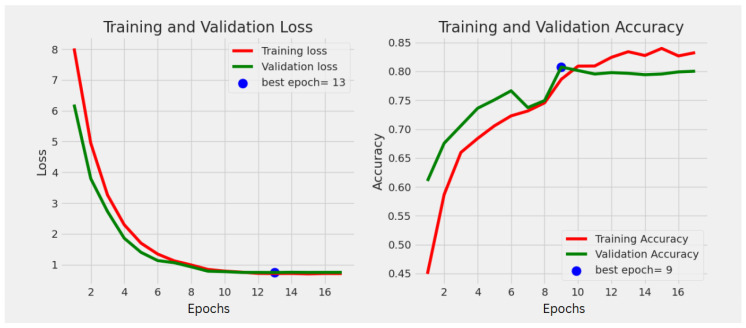
Training and validation accuracy (top) and loss (bottom) curves for the final EfficientNetB5 model. Validation loss reached its minimum at epoch 13 (0.67), and maximum validation accuracy was 82.00% (epoch 9). Training duration was 51 min.

**Figure 10 diagnostics-15-01332-f010:**
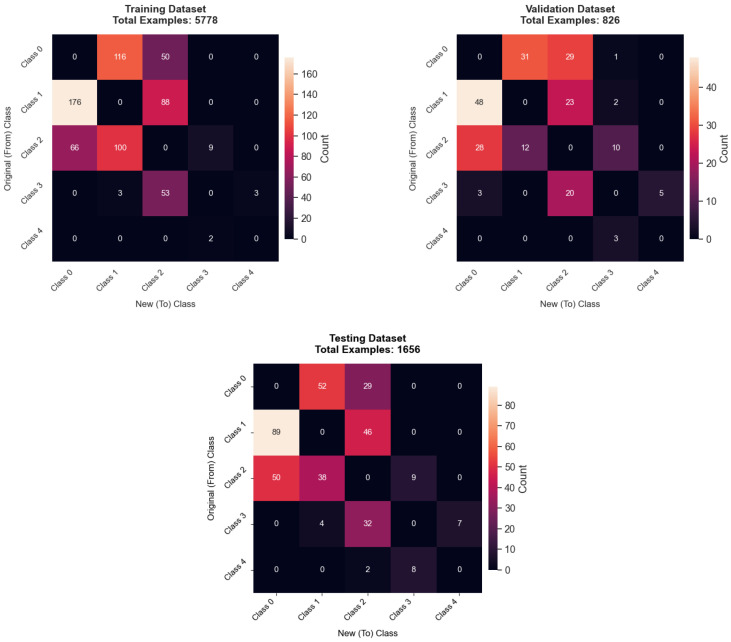
Heatmaps illustrating Cleanlab relabeling changes in the Testing (**left**), Validation (**center**), and Training (**right**) datasets. Axes represent Original (From) Class versus New (To) Class, with cell counts indicating the number of images moved between classes during the Cleanlab process.

**Figure 11 diagnostics-15-01332-f011:**
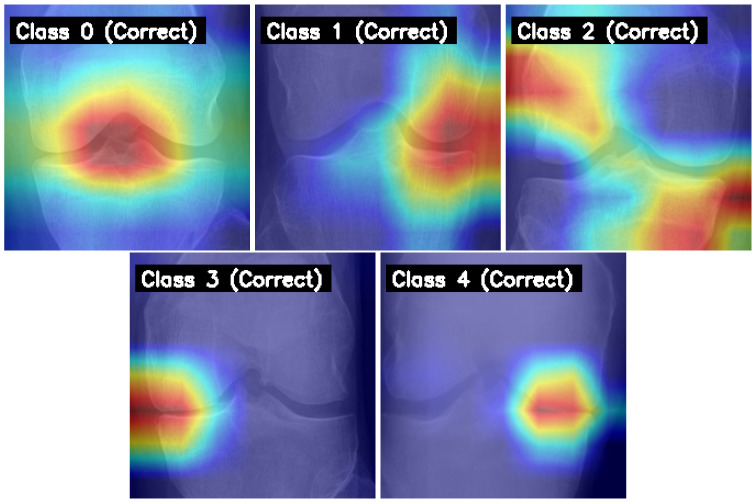
Grad-CAM visualizations for correctly predicted KOA severity grades 0 through 4 (left to right) by the final model. Red-highlighted areas indicate regions of model focus.

**Figure 12 diagnostics-15-01332-f012:**
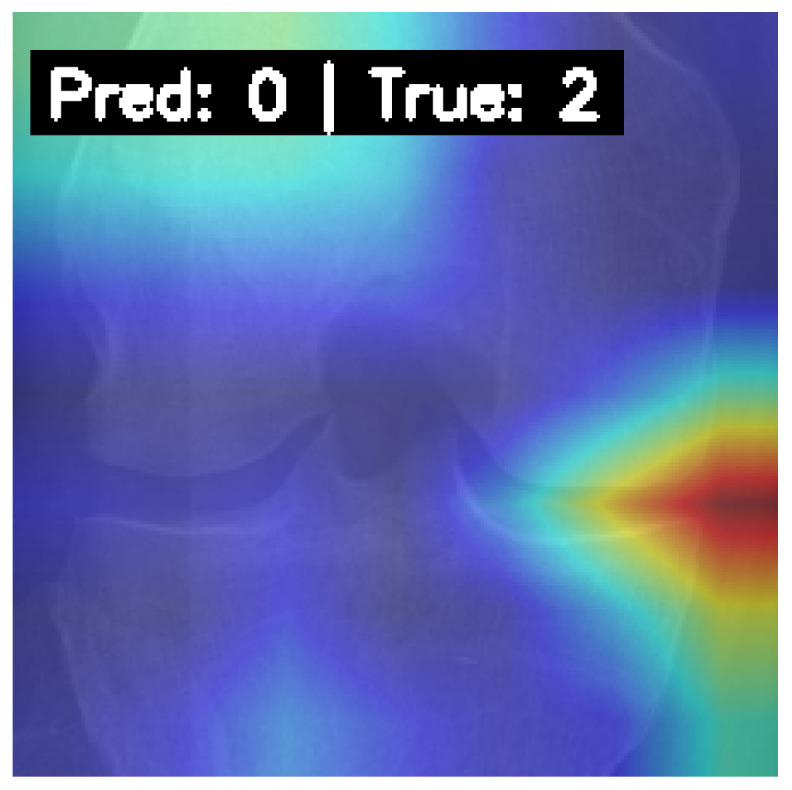
Grad-CAM visualization for a misclassified case by the final model: True Grade 2, Predicted Grade 1. The model’s attention is near the joint but appears less focused.

**Table 1 diagnostics-15-01332-t001:** Summary of models for knee osteoarthritis classification from the literature review.

Study	Model Type	Accuracy	Classes	Dataset Notes
[8]	CNN	71%	5	40,000 images (OAI dataset); compared against radiologists.
[10]	DenseNet-201	82.48%	2	5478 images (Kaggle dataset) [16]; normal vs. osteoarthritis classification.
[14]	MobileNetV3 Large	83%	2	3836 images (Kaggle dataset) [16]; optimized for constrained devices.
[11]	ResNet101	69%	5	9786 images (Kaggle dataset) [16].
[13]	VGG-16	92%	2	3836 images (Kaggle dataset) [16]; high accuracy with denoising and enhancement.
[12]	Xception	67.8%	5	8260 preprocessed images (Kaggle dataset) [16].
[15]	EfficientNet B5	97%	3	Custom layers added; 1500 images.

**Table 2 diagnostics-15-01332-t002:** Classification report for KOA grading on the Cleanlab-remediated test set (final model).

Class/Average	Precision	Recall	F1-Score	Support (# of Samples)
0 (Healthy)	0.8492	0.9369	0.8909	697
1 (Doubtful)	0.6286	0.5176	0.5677	255
2 (Minimal)	0.8432	0.7734	0.8068	459
3 (Moderate)	0.8585	0.8934	0.8756	197
4 (Severe)	0.8431	0.8958	0.8687	48
Macro Avg	0.8045	0.8034	0.8019	1656
Weighted Avg	0.8145	0.8207	0.8154	1656

Overall Accuracy: 82.07%. # refers to number of samples.

**Table 3 diagnostics-15-01332-t003:** Ablation study: impact of the full data-centric pipeline on KOA classification. Arrows in the column headers indicate whether a higher (↑) or lower (↓) value is desirable.

Configuration	Accuracy (%) ↑	Macro Prec. (%) ↑	Macro Recall (%) ↑	Macro F1 (%) ↑
Baseline (trained and evaluated on original labels)	66.36	68.36	57.87	60.23
Final pipeline (trained and evaluated on CleanLab labels)	**82.07**	**80.45**	**80.34**	**80.19**

Note: The model used to generate the confident-joint for CleanLab was first trained on the outlier-filtered set; the final pipeline model was retrained from scratch on the fully relabeled dataset and evaluated on the relabeled test set.

**Table 4 diagnostics-15-01332-t004:** Comparison of 5-class KOA classification models.

Study	Model Type	Accuracy (%)	Classes	Dataset Notes
[8]	CNN	71	5	40,000 images (OAI dataset)
[11]	MobileNetV2	67	5	9786 images (Kaggle [16])
[11]	ResNet101	69	5	9786 images (Kaggle [16])
[11]	VGG16	66	5	9786 images (Kaggle [16])
[11]	VGG19	64	5	9786 images (Kaggle [16])
[11]	InceptionResNetV2	63	5	9786 images (Kaggle [16])
[11]	DenseNet121	64	5	9786 images (Kaggle [16])
[12]	Xception	67.8	5	8260 preprocessed images (Kaggle [16])
**Our Approach (Final Pipeline)**	**EfficientNetB5 + Relabeled Dataset)**	**82.07**	**5**	**Cleanlab-remediated Kaggle [16]**

Note: Results from previous studies are as reported in their respective publications for 5-class KOA severity on the specified datasets. Our approach’s accuracy is on the Cleanlab-remediated version of the Kaggle dataset.

## Data Availability

Data can be found at https://www.kaggle.com/datasets/shashwatwork/knee-osteoarthritis-dataset-with-severity (accessed on 26 December 2024).

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
