# Peer review of "Optimizing CNN-Based Diagnosis of Knee Osteoarthritis: Enhancing Model Accuracy with CleanLab Relabeling"

_diagnostics, 2025, doi:10.3390/diagnostics15111332_

Round 1

Reviewer 1 Report

Comments and Suggestions for Authors

This manuscript investigates the performance of the EfficientNetB5 model in classifying KOA severity across five distinct classes defined by the Kellgren-Lawrence grading system, utilizing a dataset that was preprocessed and relabeled with Cleanlab to address potential label inconsistencies.

My concerns are as follows:

  1. The contributions of these manuscript should be listed in the Introduction section.
  2. The literature review lacks depth. Authors should improve the literature review Sub-sections should be added and make a review about one aspect related to this study in each sub-section. At present, the literature review just listed several works addressing the same problem, and each work corresponds to a paragraph.
  3. The novelty of this manuscript is limited. In terms of the deep learning models, this manuscript only used the existing models(e.g. EfficientNetB5), but lacks improvements and innovative approaches to the models. In terms of the analysis of the medical experimental results, there is no integration of clinical perspectives to interpret the experimental findings.
  4. The texts in Figure 6 is too small to see clearly.
  5. In table 3, is the experimental results of compared methods produced by the author by running the corresponding model codes, or copied directly from the referenced study [7] and [10]?
  6. Authors cancompare/comment on the computational complexity of the compared methods.
  7. In the tables ofresults, authors need to include up-arrow or down-arrow next to each evaluation metrics, to show whether higher or lower value is better.
Comments on the Quality of English Language

The English could be improved to more clearly express the research.

Author Response

As an addition to Comment 3 in regard to Integration of Clinical Perspectives:
In the current study, direct manual review of all Cleanlab-corrected labels by a panel of clinical experts was not performed due to resource constraints at our institution. However, we acknowledge this as a limitation and an important direction for future work.

Reviewer 2 Report

Comments and Suggestions for Authors

This paper proposes a strong pipeline using EfficientNetB5 combined with CleanLab relabeling to classify knee osteoarthritis (KOA) severity across five classes based on the Kellgren-Lawrence grading.

The paper is promising, but some points should be clarified or improved:

  1. Did you consider using class-weighted loss functions or oversampling techniques during training to better balance model attention across classes?
  2. Did clinical experts manually review any corrected labels to confirm that CleanLab improved label quality?
  3. Adding Grad-CAM visualizations of what EfficientNetB5 focuses on would help build trust, especially for clinical adoption.
  4. The authors could further strengthen their literature review by discussing recent advancements in federated learning for medical imaging. This is relevant because federated learning can help train robust KOA classification models across institutions without compromising patient data privacy, addressing the data scarcity and generalization issues noted in the paper.
  5. Have you considered validating the trained model on external datasets like the full Osteoarthritis Initiative (OAI) to test generalization?
  6. Line 28: "...subjective, leading to inconsistencies among radiologists." – good point, but you could briefly mention how inter-rater variability rates (e.g., kappa values) usually are for KOA grading to strengthen the argument.

Reviewer 3 Report

Comments and Suggestions for Authors

Reviewer 4 Report

Comments and Suggestions for Authors

The topic is justified. The paper presents an approach using EfficientNetB5 and CleanLab for KOA severity classification. It is suitable for publication after the following issues are addressed:

  • Add complete details of the dataset and preprocessing steps.
  • Replace placeholder citations and enhance the reference list.
  • Include quantitative/statistical validation of performance metrics.
  • Clarify CleanLab’s operational integration and thresholds used.

Detailed remarks to improve the draft.

  1. ABSTRACT: Minor grammatical issues are present (e.g., “an accuracy of 82.07” should be “an accuracy of 82.07%”). The abstract could be improved by specifying the dataset size and the nature of preprocessing for transparency.
  2. Overall, several minor typos and grammatical issues are scattered throughout the text.
  3. Introduction section: The Kellgren-Lawrence (KL) system is introduced, but lacks a proper citation ("[? ]" is a placeholder). There's redundancy in explaining CNNs and transfer learning, which could be condensed for better flow.
  4. Methodology: There is insufficient detail on the dataset: sample size per class, total images, and data split strategy are missing. CleanLab’s integration process (e.g., which metrics or thresholds were used to relabel data) is not described in enough technical depth. The absence of statistical significance testing (e.g., p-values, confidence intervals) limits the robustness of reported improvements.
  5. Experiments and Results: Lack of comparison against more state-of-the-art models beyond MobileNetV3 and Xception is a missed opportunity. Absence of a detailed ablation study to isolate CleanLab’s exact impact on performance.
  6. Discussion and Conclusions: The broader implications for clinical integration are briefly mentioned but not critically discussed. Future work could be more clearly defined (e.g., how to handle ambiguous classes or integrate multimodal data).
Comments on the Quality of English Language

several minor typos and grammatical issues are scattered throughout the text.

Round 2

Reviewer 1 Report

Comments and Suggestions for Authors

The authors have modified the manuscript according to the suggestions one by one, including that about the contribution list, literature review, experimental details, and ect.